# *Toxoplasma gondii* infection and high levels of IgE are associated to erythema nodosum leprosy (ENL)

**Leticia Silva Nascimento**[1☉], **Yuri Scheidegger de Castro**[1☉], **Jessany de Aquino Figueira**[1], **Rebeka da Conceição Souza**[1], **Juliana Azevedo da Silva**[1], **Edilbert Pellegrini Nahn, Júnior**[2,3], **Alba Lucínia Peixoto-Rangel**[1]*

1 Laboratório de Biologia do Reconhecer, Centro de Biociências e Biotecnologia, Universidade Estadual do Norte Fluminense Darcy Ribeiro, Campos dos Goytacazes, Rio de Janeiro, Brazil, 2 Faculdade de Medicina de Campos, Campos dos Goytacazes, Rio de Janeiro, Brazil, 3 Universidade Federal do Rio de Janeiro, Macaé, Rio de Janeiro, Brazil

☉ These authors contributed equally to this work.

* alba@uenf.br

**Data Availability Statement:** All relevant data are within the manuscript and its Supporting information files.

## Abstract

Leprosy is a chronic infectious disease caused by the bacillus *Mycobacterium leprae*. The disease may evolve for inflammatory reactions, reversal reaction (RR) and erythema nodosum leprosum (ENL), the major cause of irreversible neuropathy in leprosy, which occur in 1 in 3 people with leprosy, even with effective treatment of *M. leprae*. Leprosy remains persistently endemic in our region where it predominantly affects lowest socioeconomic conditions people, as *Toxoplasma gondii* infection in the municipality studied. Previously, we have shown *T. gondii* coinfection as a risk marker for leprosy, mainly in its severe form. This present study assessed whether *T. gondii* infection is also a risk factor for leprosy reactions and the predictive value of immunoglobulin production prior to development of leprosy reactions. Patients with leprosy (n = 180), co-infected or not with *T. gondii*, had their serum investigated for levels of IgA, IgE, IgG1, IgG2, IgG3 and IgG4 anti-PGL-1 by ELISA prior to development of leprosy reactions. The serologic prevalence for *T. gondii* infection was 87.7% in leprosy reaction patients reaching 90.9% in those with ENL. The leprosy reaction risk increased in *T. gondii* seropositive individuals was two-fold ([OR] = 2.366; 95% confidence interval [CI 95%]: 1.024–5.469) higher than those seronegative, and considering the risk of ENL, this increase was even more evident (OR = 6.753; 95% CI: 1.050–72.85) in coinfected individuals. When evaluated the prediction of anti-PGL-1 immunoglobulin levels for development of leprosy reactions in patients coinfected or not with *T. gondii*, only the increase IgE levels were associated to occurrence of reactional episodes of leprosy, specifically ENL type, in patients coinfected with *T. gondii*, compared to those not coinfected or no reaction. Thus, the immunomodulation in co-parasitism *T. gondii*–*M. leprae* suggest increased levels of IgE as a biomarker for early detection of these acute inflammatory episodes and thereby help prevent permanent neuropathy and disability in leprosy patients.

**Funding:** This project has been supported by Foundation Carlos Chagas Filho Research Support of the State of Rio de Janeiro (FAPERJ) - APQ-1 E-26/111.196/2014. This study was also financed in part by the Coordenação de Aperfeiçoamento de Pessoal de Nível Superior – Brasil (CAPES) – Finance Code 001.

## Introduction

Leprosy is a chronic disease caused by the bacillus *Mycobacterium leprae* [1]. Despite decades of programs using multidrug therapy (MDT), leprosy remains persistently endemic in many regions in tropical and subtropical underdeveloped countries [2]. The broad spectrum of clinical and histopathological manifestations of leprosy is due to the diversity of the immune response developed against *M. leprae* [3]. Both innate and acquired immune responses are involved in defense against the pathogen. The response polarization for cellular or humoral specific immunity to *M. leprae* is considered of great importance in the determination of the clinical form [3].

Some individuals present immunological instability to the antigens of the *Mycobacterium leprae* developing leprosy reaction presentment, characterized by an acute inflammation in the course of the disease. These episodes occur mainly in patients with the multibacillary form, and may occur before, during or after WHO standard multidrug therapy (MDT) [4]. Leprosy reactions are frequent immune-mediated complications that occur in 30 to 50% of patients with leprosy and can cause high morbidity [5]. They are divided into type 1 reaction, or Reversal Reaction (RR), and type 2 reaction, or Erythema Nodosum Leprosum (ENL), each one presenting specificity inherent to the pathophysiology, clinical picture and therapy [6].

The morbidity of leprosy reactions occurs mainly because of the acute and aggressive involvement of the peripheral nerves. The neurological inflammatory reaction can lead to permanent changes in the functions of the compromised nerves, called neurological sequels of leprosy. Clinically, the sequelae present themselves in different ways, such as chronic neuropathic pain, incapacitating paresis or physical deformities. Both reactional episodes and neurological sequelae stigmatize patients with the disease and are associated with impaired quality of life [7].

The precise mechanisms that trigger leprosy reactions are still unknown. However, a series of clinical variables were associated, to a greater or lesser extent, with its occurrence, with some controversies persisting. Factors such as pregnancy, infection, vaccination and psychological stress, increasing age and illness with high disability have already been considered to cause reactional states, however, these associations have not been confirmed [8, 9].

According to data from previous studies, infection by *T. gondii* presents as a risk factor for the development of leprosy, especially in its most severe form (Lepromatous) that a Th2 immunological response predominates. Proliferation of the bacillus occurs, with the occurrence of many lesions and disseminated infiltrations in the skin and nerves. [10]. Are recognized that reactional episodes of leprosy are closely linked to morbidity and physical disability associated with disease progression [8] especially in patients with the multibacillary form. Since infection by *T. gondii* is a risk factor in the development of leprosy, mainly in its severe form, it is pertinent to know whethe*r T. gondii* infection is also related with disease progression to leprosy reactions and physical disabilities.

Toxoplasmosis is a disease caused by the protozoan apicomplexa *Toxoplasma gondii*, an obligate intracellular parasite with cosmopolitan distribution that infects warm-blooded animals [11]. Is one of the zoonoses that most affects humans worldwide and in the vast majority of the cases manifests itself as asymptomatic due to the immunocompetence of individuals [12].

The prevalence of *T. gondii* infection in Brazilian adults can vary from 50% to 80% [13]. In Campos dos Goytacazes—RJ, studies were carried out with a sample of 367 individuals where 65.9% of these showed positive serology for *T. gondii* [14].

Immunocompetent hosts have an excellent cellular immune response against infection by this protozoan leading to the occurrence of lysis of infected cells and inhibition of their multiplication. The CD4+ and CD8+ T cells, responsible for protective immunity, play a crucial role

in controlling the infection in the acute phase, while the pro-inflammatory cytokines IL-12, IFN- γ and TNF-α confer resistance to the parasite [15].

Immunoglobulin enables antibodies to enhance their effector function, thus contributing to the diversity of the immune response.

In this work, we evaluated the co-parasitism *M. leprae*—*T. gondii* with the occurrence of reactional episodes in patients before initiating multidrug therapy, identifying parameters of the immune response that might be associated with the occurrence of these reactional episodes.

## Materials and methods

### Participants and recruitment

Patients were recruited on a voluntary basis between October 10, 2019 and August 7, 2023 in a Reference Centre for Leprosy, Campos dos Goytacazes city, Rio de Janeiro. Leprosy was diagnosed based on clinical and bacteriological observations and classified according to Madrid Classification. The Madrid classification is based on clinical and bacilloscopic characteristics, dividing leprosy into two unstable groups, indeterminate and dimorphic or borderline, and two stable types, tuberculoid and lepromatous polar [16].

Participant information was collected with emphasis on standardizing data collection and definition of reactions between all patients. Newly diagnosed, untreated leprosy patients without clinical reactions were enrolled and blood was drawn before initiation of MDT (n = 180) (Fig 1), since MDT administration could influence with immunological parameters. Clinical monitoring for reactions was performed during the therapy and after completion MDT. Patients presenting more than one type of reaction simultaneously were also recruited (n = 6). Patients diagnosed with reactions at the first clinic visit (n = 6), before initiation of MDT, were also recruited and their blood samples was drawn. Patients with leprosy relapse and neural leprosy were excluded from the analysis due to there is no consensus regarding the classification of the neural form. In 1952, Wade [17] classified it as an independent subgroup of leprosy, but this generated many discussions, as some authors believe that it is part of the subgroups of the disease. This form does not fit into any of the subgroups, so it must be analyzed separately, as it presents a difficult diagnosis, the clinical manifestations differ from the other forms, as it does not present skin lesions and the bacilloscopy is negative.

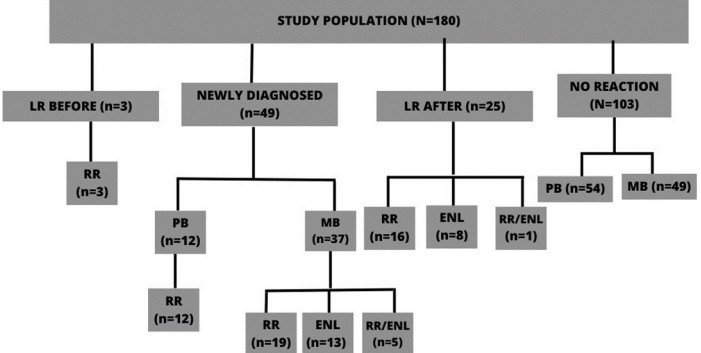

**Fig 1. Flow chart of study.** LR: Leprosy Reaction; PB: paucibacillary; MB: multibacillary; RR: reversal reaction; ENL: erythema nodosum leprosy; RR/ENL two reactions at the same time; LR Before: patients diagnosed with reactions at the first clinic visit; LR After: patients that developed reaction after follow-up.

**Table 1. Number of patients according to the analyzes.**

| VARIABLES | LEPROSY WITH REACTION | NO REACTION | TOTAL |
|---|---|---|---|
| **ANALYZES** | | | |
| Detection of anti-soluble toxoplasma antigen (STAg) antibodies | 77 | 103 | 180 |
| ELISA of immunoglobulins: anti PGL1 IgA, IgE, IgG1, IgG2, IgG3 and IgG4 | 42 | 52 | 94 |

*Due to experimentation problems, the number of individuals analyzed were different.

In brief, newly diagnosed leprosy cases without RR or ENL were followed for 3 years, and then grouped by whether or not they developed RR, ENL or both (RR/ENL) during follow-up: 1.) No reaction; 2.) Developed RR (RR); 3.) Developed ENL (ENL); 4) Developed RR and ENL simultaneously (RR/ENL).

From 180 patients, 77 patients developed leprosy reaction and 103 not developed leprosy reaction. All of these patients were analyzed for the detection of anti-STAg (anti- Soluble *Toxoplasma gondii* Antigen) IgG antibodies. Of 180 patients, 94 were analyzed regarding the immunoglobulin levels (IgA, IgE, IgG1, IgG2, IgG3 and IgG4) against anti-PGL-1 from *M. leprae* (Table 1). The difference in the number of samples between the tests was due to the insufficient amount of reagents (antibodies) available to carry out the analysis of Igs anti-PGL-1 in all samples. The choice of samples was random for each clinical group.

## Ethics

This study was performed according to the Helsinki Declaration. Written informed consent was obtained before enrolment. Patients received treatment according to national guidelines. Ethical approval of the study-protocol was obtained through Faculdade de Medicina de Campos Human Ethical Research Committee (CAEE No. 19679119.8.0000.5244).

## Soluble *Toxoplasma gondii* antigen preparation

Tachyzoite forms of *Toxoplasma gondii* parasites RH strain, maintained in female swiss mice, of approximately, 3–4 weeks old were recovered 2–3 days after infection, for Soluble *Toxoplasma gondii* Antigen (STAg) production [10]. The animals were euthanized using an excessive dose of anesthetic Urethane and peritoneal fluid containing *Toxoplasma gondii* tachyzoites was obtained by peritoneal lavage. Peritoneal fluid was centrifuged at $100 \times g$ for 5 min. the supernatant was centrifuged again at $913 \times g$ for 30 min at 4°C. A small quantity of PBS was added to the parasite sediment and an aliquot of this suspension was removed and diluted 1:100 for counting in a Newbauer chamber. Approximately, $2.5 \times 10^8$ parasites per milliliter were exposed to six pulses of 30 s each, in ice, using ultrasound equipment (Branson–Sonifier 150) and centrifuged at $900 \times g$ for 20 min [10]. The supernatant was transferred to another tube and centrifuged again at $10000 \times g$ for 10 min. Protein concentration in the supernatant (STAg) was determined by Lowry method [18], and the antigen was then stored at -20°C until use. Animal handling was conducted in accordance with the guidelines of the National Council for the Control of Animal Experimentation and international recommendation [19]. Experimental protocols for STAg production were approved by Animal Use Ethics Committee from State University of Northern Rio de Janeiro under protocol number 385.

## Enzyme linked immunosorbent assay (ELISA) for detection of anti-soluble toxoplasma antigen (STAg) antibodies

A 96-well microtiter plates (Nunc™ *MaxiSorp*™) were coated with 100 μl of 0.1 mol/L bicarbonate buffer, pH 9.6, containing STAg (10 μg/ml), for 18–20 h at 4˚C as previously described by Carvalho et al., 2008 [20], with some modifications. The plates were three times washed with PBST (PBS [Phosphate Buffered Saline] 1×; 0.05% Tween 20). Then, 100 μl of blocking buffer (PBST, 1% BSA [Bovine Serum Albumin]) was added in each well and incubated for 30 min at 4˚C. The plates were washed three times and samples (including positive and negative controls) were diluted 1:1000 in diluent buffer (PBST; 0.5% BSA) and added 100 μl to the wells in duplicated. The samples were incubated for 1 h at room temperature. After washing, antibodies IgG HRP anti-human (Southern Biotech, Alabama, USA) were diluted 1:1000 in diluent buffer and 100 μl added to each well and incubated for 1 h at room temperature. The plates were washed and 100 μl of a freshly prepared substrate solution (28 mmol/L citric acid, 48 mmol/L dehydrated sodium phosphate, 1 mg/ml ABTS [2,2'-AZINO-BIS(3-ETHYLBEN-ZOTHIAZOLINE-6-SULFONIC ACID) DIAMMONIUM SALT] and 0.003% $H_2O_2$) was added for color development. The reaction was stopped by the addition of 30 μl of citric acid (0.2 mol/L) and plates read at 405 nm in a reader (VersaMax™ Tunable Microplate, VWR International, Pensilvânia, USA).

The cut-off point of the test was calculated by the mean of negative controls plus three times the standard deviation of these samples, where values below or equal the cut-off were considered negative, and values above the cutoff were considered positive.

## Enzyme linked immunosorbent assay (ELISA) of immunoglobulins: IgA, IgE, IgG1, IgG2, IgG3 and IgG4

IgA, IgE, IgG1, IgG2, IgG3 and IgG4 anti-PGL-1 antibodies were detected by enzyme-linked immunosorbent assay (ELISA) as previously described by Bazan-Furini [21], with some modifications. 96-well ELISA plates were primed with the following reagent kindly donated by BEI Resources, NIAID, NIH: *Mycobacterium leprae* phenolic glycolipid-1 (PGL-1), (NR-19342) at a concentration of 2.0 μg/mL per well in 100 μL carbonate/bicarbonate buffer 0 1M (pH 9.6) for 18–20 hours at 4˚C. After sensitization, plasma from leprosy patients and controls were added in duplicates, diluted 1:10 in 100 μL Tris/Tween buffer 0.05% 15mM (pH 7.5) containing 5.0% bovine serum albumin (BSA–Sigma Aldrich), and incubated for 2 hours at 37˚C. After the incubation period, anti-immunoglobulin antibodies (IgG1, IgG2, IgG3 and IgG4 – SoutherBiotec) were diluted in 100μL of Tris/Tween 0.05% 15mM (pH 7.5) containing 5.0% BSA, and the plates were again incubated for 2 hours at 37˚C. Subsequently, three washes were performed with Tris/Tween 0.05% 15mM (pH 7.5), and streptavidin-peroxidase diluted 1:2500 in 100μL of Tris/Tween 0.05% 15mM was added (pH 7.5) containing 5.0% BSA, and the plate was incubated for 30 min at 37˚C. After this period, the wells were washed three times with Tris/Tween 0.05% 15mM (pH 7.5) and ABTS substrate solution (2,2' Azinobis acid; 3-ethylbenzothiazolin-6-sulfonic acid) diammonium salt (Sigma -Aldrich) was added to reveal and kept in the dark for 15 to 20 minutes. The plates were read using a plate reader (EPOCH--BioTek) and the 405nm filter was used.

## Statistical analysis

Statistical analyses used a significance level of 5% and were performed using GraphPad Prism v.6 Software (GraphPad Software, La Jolla, CA). To evaluate differences in immunoglobulins production in leprosy reaction patients coinfected or not with *T. gondii*, the nonparametric

test Kruskal-Wallis, followed by Dunn's test (posttest), was applied for comparisons among three or more groups. While the nonparametric Mann-Whitney test was applied for comparison between two groups. For the analysis of anti-PGL-1 immunoglobulins, the average of each patient were divided by the average of the blank of each ELISA plate in order to get an index value. The index values were statistically analyzed among groups. The odds ratio with confidence interval of 95% analyzes were also performed using Prism with contingency tables and Fisher's test application.

## Results

### Serological prevalence of *T. gondii* infection in followed up leprosy patients

The prevalence of IgG seropositivity against *T. gondii* was increased in leprosy patients that developed leprosy reactions (68/88.3%) compared to those not develop leprosy reactions (9/11.6%) Table 2. Considering the types of leprosy reactions, the prevalence for toxoplasmosis was 83% (42) in patients with RR, 90.9% (20) in patients with type 2 ENL and 100% (6) in patients with both reactions (RR/ENL). These data indicate a high prevalence of *T. gondii* infection in patients that developed leprosy reactions.

### Association between *Toxoplasma gondii* infection and development of leprosy reactions

Leprosy patients coinfected with *Toxoplasma gondii* had twice the risk of developing leprosy reactions compared to those not coinfected Table 3. Additionally, when stratifying the types of leprosy reactions, *T. gondii* infection increased in six-fold the risk for ENL (OR = 6.753; 95% CI: 1.050–72.85), compared to RR and RR/ENL that not present statistically significant Table 2. This data suggests *T. gondii* infection as a risk factor for development of ENL in leprosy patients.

### Predictive value of anti-PGL-1 immunoglobulins for leprosy reaction in leprosy patients serum

The antibodies (IgA, IgE, IgG1, IgG2, IgG3 and IgG4) produced among leprosy patients coinfected or not with *T. gondii* were evaluated to identify whether there was differential production of these immunoglobulins of predictive value for development of leprosy reaction (Fig 2).

There was no significative differences in the production of IgA, IgG1, IgG2, IgG3 and IgG4 between who developed leprosy reaction and those who did not develop a leprosy reaction (Fig 2). Nonetheless, leprosy patients who developed leprosy reaction and were also coinfected with *T. gondii* had elevated levels of IgE (Fig 2b) compared to individuals without reaction and

**Table 2. Serological prevalence for *Toxoplasma gondii* infection in leprosy patients prior the development of leprosy reactions.**

| Individuals | Clinical Groups | Nº | +IgG anti-*T. gondii* n (%) | - IgG anti -*T. gondii* n (%) |
|---|---|---|---|---|
| **Leprosy Reaction** | RR | 50 | 42 (83%) | 8 (17%) |
| | ENL | 21 | 20 (90.9%) | 1 (9.1%) |
| | RR/ENL | 6 | 6 (100%) | 0 (0%) |
| **No Leprosy Reaction** | | 103 | 77 (74.8%) | 26 (25.2%) |
| **Total** | | | 145 (80.5%) | 35(19.4%) |

*n+ patients co-infected with *T.gondii* and n- patients infected only with *M. leprae*, RR: reversal reaction; ENL: Erythema Nodosum Leprosum (ENL); RR/ENL two reactions at the same time.

**Table 3. *Toxoplasma gondii* infection and risk analysis for leprosy reaction development.**

|  | *T. gondii* positive | *T. gondii* negative | OR | 95% CI | P |
|---|---|---|---|---|---|
| Leprosy Reaction | 68 | 9 | 2.355 | 1.044–5.656 | **0.0400** * |
| Leprosy No Reaction | 77 | 26 |  |  |  |
| RR | 42 | 8 | 1.773 | 0.7328–4.088 | 0.1971 |
| Leprosy No Reaction | 77 | 26 |  |  |  |
| ENL | 20 | 1 | 6.753 | 1.050–72.85 | **0.0382** * |
| Leprosy No Reaction | 77 | 26 |  |  |  |
| RR/ENL | 6 | 0 | 2.423 | 0.3959–28.21 | 0.4040 |
| Leprosy No Reaction | 77 | 26 |  |  |  |

(OR) Odds ratio; (CI) Confidence Interval; (RR) Reversal Reaction; (ENL) Erythema Nodosum Leprosum; (RR/ENL) two reactions at the same time. Fisher's test for analyzes;

*Statistical Significance

not co-infected *(p = 0.0141)*, without reaction coinfected *(p = 0.0083)* and with reaction not coinfected *(p = 0.0417)*, suggesting that the immunomodulation caused by coinfection with *T. gondii* can increase IgE levels and lead to the occurrence of leprosy reactional episodes.

To evaluate whether this increase in IgE levels could be associated to the development of a specific type of leprosy reaction, we measured the levels of IgE in patients who developed RR and ENL coinfected or not with *T. gondii*. We found that anti-PGL-1 IgE levels were increased in ENL patients, co-infected with *T. gondii*, compared to those with reaction not coinfected *(p = 0.0132)* and without reaction coinfected *(p = 0.0175)* or not infected *(p = 0.0132)*. This data suggests that *T. gondii* can be influencing the modulation of the immune response in ENL coinfected patients by increasing the production of IgE compared to RR coinfected patients or RR and ENL not coinfected patients (Fig 3).

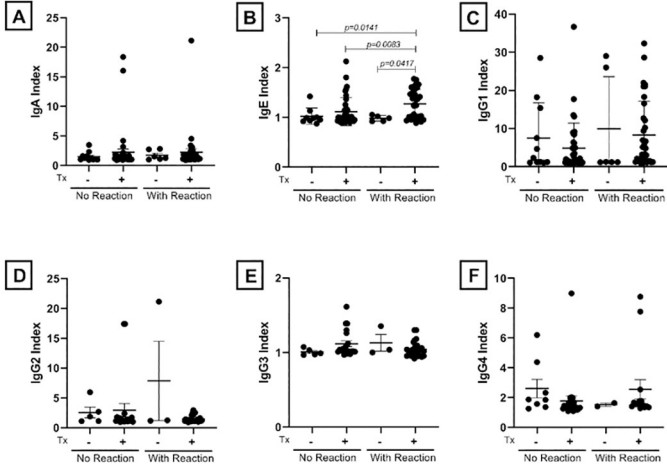

**Fig 2. Immunoglobulins anti-PGL-1 present in the serum of leprosy patients coinfected (Tx+) or not (Tx -) with *T. gondii*, who developed (with reaction) or not (without reaction) leprosy reactions.** The IgA levels (A), IgE (B), IgG1 (C), IgG2 (D), IgG3 (E) and IgG4 (F) were measured by ELISA and are presented as index value. The average of each patient were divided by the average of the blank of each ELISA plate in order to get an index value. The groups were compared using Kruskal-Wallis followed by Dunn's statistical test. The horizontal bars represent mean value and the vertical bars the standard error of the mean (SEM).

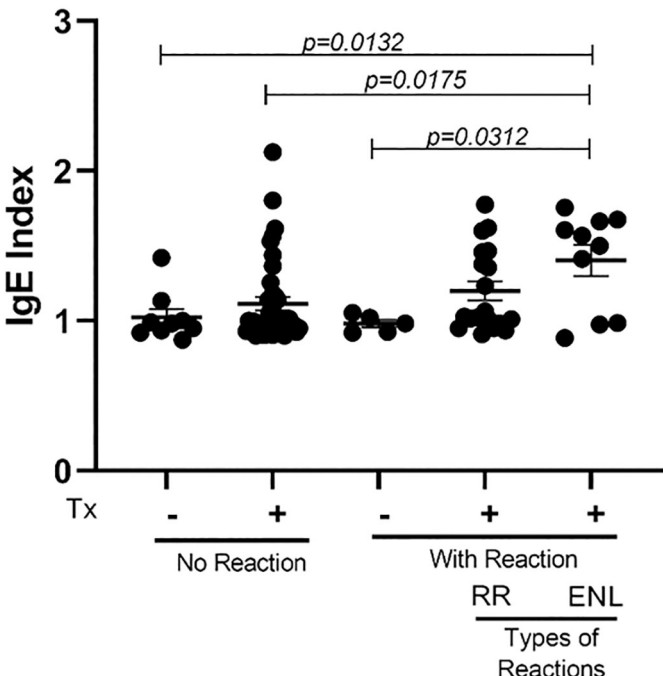

**Fig 3. Anti-PGL-1 IgE production from followed up leprosy patients who developed reversal reaction (RR) and erythema nodosum leprosum (ENL), coinfected (Tx +) or not infected (Tx -) with *T. gondii*.** The groups were compared using Kruskal-Wallis followed by Dunn's statistical test. The horizontal bars represent mean value and the vertical bars the standard error of the mean (SEM).

## Discussion

With the arrival of new scientific methodologies to identify and diagnose known and emerging pathogens, it is clear that co-infections are a common phenomenon. Co-infection with more than one pathogen, such as HIV, *Mycobacterium tuberculosis*, hepatitis virus, helminths, and *Plasmodium*, is estimated to affect about one-third of the human population in developing countries [22]. Epidemiological data suggest a higher incidence of negative effects on pathogen-specific host immune responses during co-infection [23]. However, the underlying mechanisms remain poorly understood [22]. However, epidemiological evidence suggests that many chronic infections may increase susceptibility and pathology induced by unrelated pathogens [22].

In the present study, we hypothesize that *T gondii* infection might be a risk factor also for development of leprosy reactions in leprosy patients, since the most cases of inflammatory reactions affect multibacillary patients. Therefore, anti-STAg IgG serological analyzes revealed a high prevalence of *T. gondii* infection in leprosy patients who developed leprosy reactions (87.7%). When stratifying the patients in according to the types of reaction, the prevalence of *T. gondii* infection reach 90.9% in patients who developed ENL. The correlation between the *T. gondii* coinfection with the occurrence of leprosy reactional episodes (RR, ENL and RR/ENL) was shown to be two-fold increased risk factor (OR = 2.355; 95% CI:1044–5656) compared to seronegative individuals. When we stratified for the types of reaction, it was possible to observe a six-fold increased risk (OR = 6.753; 95% CI:1.050–72.850) for the development of ENL reaction. Contrary to our data, BALB/c mice chronically infected with the intracellular protozoan *T. gondii* or *Besnoitia jellisoni* were resistant to footpad challenge with *M. leprae*.

Resistance was manifested by lower numbers of recoverable *M. leprae* in the footpads of protozoal-infected mice and resistance was enhanced in *Toxoplasma*-infected mice by a booster injection of *Toxoplasma* antigen in the infected footpad [24]. However, murine is not a good model to study the leprosy infection, since rodents do not systemically develop leprosy disease. There are no studies in humans about immune response modulation in patients co-infected with *M. leprae* and *T. gondii* and the influence of this immunomodulation on the clinical manifestation of leprosy and toxoplasmosis symptoms. These infections have an opposite protection immune response, where toxoplasmosis majority elicits a Th1 cellular immunity, which induces the production of IL-12, IL-2, IFN-α and TNF-α cytokines; while individuals with the most severe form of leprosy (lepromatous) generally develop humoral immune response (Th2 type) with production of IL-4, IL-5, IL-10 and IL-13, which suppress macrophage activities and stimulate mast cell and B lymphocyte activation. There is no clear relationship between levels of any anti-PGL-1 immunoglobulins and predicted development of a leprosy reaction. [25, 26].

Here, we have shown the enhances IgE levels in leprosy patients coinfected with *T. gondii* are associated to the occurrence of leprosy reactional episodes. Besides its critical role in allergy and immunity towards helminthic parasites [27], *in vivo* expression of IgE has been observed during protozoal infections such as those caused by *Plasmodium* spp. [28, 29], *Leishmania* spp. [30] and *Trypanosoma cruzi* [31], although the role of this immunoglobulin in anti-microbial immunity remains unclear [32]. IgE/antigen bound to human cells through FceRI and FceRII surface molecules [33]. Macrophages, which are pivotal effectors for control of intracellular and extracellular parasites, fail to express FceRI but may bound IgE through surface FceRII/CD23 antigen [34]. Vouldoukis and colleagues (2011) [35] clearly shows that macrophages, in the absence of FceRI, express FceRII at enough surface levels that enable them to cross-link these receptors by IgE-IC or other physiologic ligands. Consistently, CD23 expression following infection with *T. gondii* parasites may be due to their ability to induce the transcription factor STAT6 [36], which is involved in the induction of Th-2 gene promotion including CD40, CD23 and IgE. Finally, high levels of TNF-a and NO mediated through IgE generation may also account for deleterious chronic inflammatory diseases that are observed during many parasitic infections and which may be similar to what happens in leprosy reactions.

The presence of other intestinal parasites, allergies and asthma should be also considered because these diseases could trigger Th-1 or Th-2 immune response components and may indirectly influence in the magnitude of immune response against *M. leprae*. Nonetheless, this does not prejudice our findings since leprosy patients with negative serology for *T. gondii* are also subject to these same poor living conditions but did not have high levels of IgE against *M. leprae* PGL-1 and association with the development of a leprosy reaction.

Our results suggest that *T. gondii* infections may play a role in the progression to more severe type of leprosy reaction, ENL, due to immunomodulation that influences the increase in IgE levels against the *M. leprae* antigen (PGL-1) in *T. gondii* coinfected patients, but not in those not coinfected. These findings could serve as a fundamental basis for clinicians to perform immunological tests for early detection of acute inflammatory episodes and thus help to prevent permanent neuropathy and disabilities in leprosy patients, since it is the greatest stigma suffered by them. However, further studies are needed to investigate how *T. gondii* could be contributing to a more severe manifestation of the leprosy reaction.

## Supporting information

**S1 File. Underlying data set.** The spreadsheet called "Immunoglobulins" refers to the index values used in analyzing the levels of anti-PGL1 immunoglobulins. The worksheets called

"Leprosy No Reaction", "Reverse Reaction". "Erythema Nodosum Leprosy" and "RR-ENL" present values used for risk analysis calculations.
(XLSX)

## Acknowledgments

We thank the Hansen Health Program and the Blood Bank (Hemocenter) of Campos dos Goytacazes for assisting with patients at the recruiting facilities. The following reagent was obtained through BEI Resources, NIAID, NIH: *Mycobacterium leprae* Phenolic Glycolipid-I (PGL-I), NR-19342.

## Author Contributions

**Conceptualization:** Alba Lucínia Peixoto-Rangel.

**Data curation:** Leticia Silva Nascimento, Yuri Scheidegger de Castro, Edilbert Pellegrini Nahn, Júnior.

**Funding acquisition:** Alba Lucínia Peixoto-Rangel.

**Investigation:** Leticia Silva Nascimento, Yuri Scheidegger de Castro, Jessany de Aquino Figueira, Rebeka da Conceição Souza, Juliana Azevedo da Silva.

**Methodology:** Leticia Silva Nascimento, Yuri Scheidegger de Castro, Alba Lucínia Peixoto-Rangel.

**Project administration:** Alba Lucínia Peixoto-Rangel.

**Resources:** Edilbert Pellegrini Nahn, Júnior, Alba Lucínia Peixoto-Rangel.

**Supervision:** Alba Lucínia Peixoto-Rangel.

**Writing – original draft:** Leticia Silva Nascimento, Yuri Scheidegger de Castro, Alba Lucínia Peixoto-Rangel.

**Writing – review & editing:** Alba Lucínia Peixoto-Rangel.

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
