## [Decision Letter · Decision Letter 0]

15 Nov 2023

PONE-D-23-32570Toxoplasma gondii infection and high levels of IgE are associated to erythema nodosum leprosy (ENL)PLOS ONE

Dear Dr. Rangel,

Thank you for submitting your manuscript to PLOS ONE. After careful consideration, we feel that it has merit but does not fully meet PLOS ONE’s publication criteria as it currently stands. Therefore, we invite you to submit a revised version of the manuscript that addresses the points raised during the review process.

Dear Dr de Castro,

We have received reviewers’ comments. Before reaching to the decision, we would like you to address all comments provided by the reviewers including why 95% CI was wide for ENL and *T. gondi*i? Will it affect the interpretation of your fining?  And also respond to the issue of *M. leprae* infection in the foot of the BALB/c mouse mention in the discussion section.

**Editor’s comment**

Line #52: what kind of bacteriological examination was made for classification assuming it is difficult to grow M. leprae in artificial culture media

How mice were handled during preparation of antigen? Were they killed? If yes describe method of killing?

During recruitment of study participants, Have you considered patients previous condition such as asthma, allergy, parasitic infection which may increase level of IgE and may affect your finings

We look forward to receiving your revised manuscript.

Kind regards,

Musa Mohammed Ali, PhD

Academic Editor

PLOS ONE

https://idpjournal.biomedcentral.com/articles/10.1186/s40249-020-0636-3

In your revision ensure you cite all your sources (including your own works), and quote or rephrase any duplicated text outside the methods section. Further consideration is dependent on these concerns being addressed.

 [This project has been supported by Foundation Carlos Chagas Filho Research Support of the State of Rio de Janeiro (FAPERJ) - APQ-1 E-26/111.196/2014. This study was also financed in part by the Coordenação de Aperfeiçoamento de Pessoal de Nível Superior – Brasil (CAPES) – Finance Code 001”].  

Additional Editor Comments:

Dear Dr de Castro,

We have received reviewers’ comments. Before reaching to the decision, we would like you to address all comments provided by the reviewers including why 95% CI was wide for ENL and T. gondii? Will it affect the interpretation of your fining? And also respond to the issue of M. leprae infection in the foot of the BALB/c mouse mention in the discussion section.

Editor’s comment

Line #52: what kind of bacteriological examination was made for classification assuming it is difficult to grow M. leprae in artificial culture media

How mice were handled during preparation of antigen? Were they killed? If yes describe method of killing?

During recruitment of study participants, Have you considered patients previous condition such as asthma, allergy, parasitic infection which may increase level of IgE and may affect your finings

Reviewers' comments:

Reviewer's Responses to Questions

**Comments to the Author**

1. Is the manuscript technically sound, and do the data support the conclusions?

Reviewer #1: Yes

Reviewer #2: Partly

2. Has the statistical analysis been performed appropriately and rigorously? 

Reviewer #1: Yes

Reviewer #2: I Don't Know

3. Have the authors made all data underlying the findings in their manuscript fully available?

Reviewer #1: Yes

Reviewer #2: No

4. Is the manuscript presented in an intelligible fashion and written in standard English?

Reviewer #1: Yes

Reviewer #2: Yes

5. Review Comments to the Author

Reviewer #1: The manuscript has addressed an important issue related to co-parasitism of T. gondii – M. leprae that suggest increased levels of IgE as a biomarker for early detection of these acute inflammatory episodes and thereby help prevent permanent neuropathy and disability in leprosy patients which is not addressed by other studies. Thus, this study will be an input for early and proper management of patients with leprosy reactions.

Introduction

• RR and ENL abbreviations page 4, paragraph 3, line 100 and 101

• Please define Virchowian clearly page 6, paragraph 1, line 122

• Repetitions of sentence so it would be better if you rewrite it. page 6, paragraph 1

• Percentages should be separated by dot instead of comma (Page 6, paragraph 3, line 138) E.g 65,9%

• Reactional episodes in patients can occur before, during and after completing MDT, but you studied the co-parasitism with the occurrence of reactional episodes in patients before initiating MDT, why you wanted to do the study on before MDT? Do you have a reason? It would be good if you elaborate it. Page 7, paragraph 2, line 147.

o Do you have a follow-up plan to study the effect of this co-parasitism with the occurrence of reactional episodes in patients during and after completion of MDT?

Methodology section :-

• Would you define the Madrid classification? Since there are different types of leprosy classification, it would be better if you make it clear (Page 7, paragraph 3, line 157)

• Neural leprosy patients were excluded from the analysis. Do you have specific reasons to exclude them from the study? (Page 7, paragraph 5, line 166)

• Please define the abbreviations:

o Correct the ENL, Page 8, Figure 2, line 173,

o anti-STAg, Page 8, Paragraph 2, line 178

• The protein concentration in the supernatant (STAg) was determined by Lowry method. Would you please explain the method and also attach it as annex (page 9, Line 201).

• Please define the abbreviations:

o PBST …Page 9, and Line 208.

o BSA… Page 9, and Line 209.

• I think Immunosorption written in the Enzyme Linked Immunosorption Assay (ELISA), should be changed to Immunosorbent assay Page 10, line 224

• Please define the abbreviation

• ABTS stands for …Page 11, Paragraph 1, and Line 242.

Results section,

Percentages should be separated by dot instead of comma.

o Page 11, Paragraph 2, line 262 and 263

o Check Table 2 summation

Make corrections, highlighted Page 13, Line 282

Discussion

• The cytokine TFN, Page 14, last paragraph, line 344

• The predictive value of anti-PGL-1 immunoglobulin levels for leprosy immune reactions is unclear. Please would you make the sentence clear? Which immunoglobulin? Are you referring to IgM or IgE levels? (Page 15, 1st paragraph, Line 347)

• Finally, high levels of TNF-a and NO mediated through IgE generation may also account for deleterious chronic inflammatory diseases that are observed during many parasitic infections as leprosy reactions. (Page 15, 1st paragraph, Line 363).

The sentence is not clear it seems that leprosy reaction is a parasitic infection, though leprosy is a bacterial infection; leprosy reaction is not! Will you please rewrite it?

Conclusion:-I think this study is the ONLY study done on the increased risk of leprosy reaction with patients that are infected with T.gondii, so do you think it is possible to make conclusions as the presence of co-infection with T.gondii as high risk to develop leprosy reactions?

Reviewer #2: The manuscript has some aspects that I would like to mention. It is well written, however it is very simple and linear in the experiments and conclusions. It does not provide new tools.

Materials and Methods.

Figure 1. The authors should explain more clearly which groups are being studied and what the abbreviations mean. Line 163-166. The paragraph should be placed above on lines 156

Figure 2. Says study flow chart. There is an error it should say Figure 1.

Table 1. It does not provide further information. What is mentioned in the table could well go in the text

Line 172. It says that not all participants were included in the laboratory studies. Describe how the selection was made, and what criteria were used.

Results

Table 2. It is poorly tabulated and confusing. See another form of presentation

Figure 2. The authors describe the results of the IGE experiment (lines 287-291), however there are other results that are not explained, such as, for example. IgG1 (values are high in both groups), IgG2 and 4 (large confidence intervals) will be that the groups correspond to a low number of patients, and IgG3 (the average value is high in both groups). All these results must be explained.

Figure 3. This is oka

Discussion

Line 329-333. The conclusion regarding M. lepra infection in the foot of the BALB/c mouse is not correct because the infection did not occur due to the immune protection of T. gondii but because the mouse is never susceptible to infection with myocbactertium leprae. To affirm that, the authors mus look for a susceptible host like the armadillo.

Comments

The authors make conclusions on a single biological marker. They should, as far as possible, associate the symptoms and clinical signs of each of the patients (Madrid Classification) with the respective markers and should monitor the marker (IgG E) over time at the time of detection and during treatment to effectively determine that immunomodulation is exerted.

6. PLOS authors have the option to publish the peer review history of their article (what does this mean?). If published, this will include your full peer review and any attached files.

Reviewer #1: No

Reviewer #2: No

---

## [Author Response · Author response to Decision Letter 0]

12 Dec 2023

Reviewer #1:

 Introduction

• RR and ENL abbreviations page 4, paragraph 3, line 100 and 101

They are divided into type 1 reaction, or Reverse Reaction (RR), and type 2 reaction, or erythema nodosum leprosy (ENL), each one presenting specificity inherent to the pathophysiology, clinical picture and therapy [6]. 

• Please define Virchowian clearly page 6, paragraph 1, line 122

Virchowian is the same of Lepromatous. It means the most severe form of leprosy. In order to standard the writing of this clinical form of leprosy, we substitute Virchowian for Lepromatous in line #115: According to data from previous studies, infection by T. gondii presents as a risk factor for the development of leprosy, especially in its most severe form (Lepromatous) that a Th2 immunological response predominates. Proliferation of the bacillus occurs, with the occurrence of many lesions and disseminated infiltrations in the skin and nerves. [10].

• Repetitions of sentence so it would be better if you rewrite it. page 6, paragraph 1

Toxoplasmosis is a disease caused by the protozoan apicomplexa Toxoplasma gondii, an obligate intracellular parasite with cosmopolitan distribution that infects warm-blooded animals [11]. Is one of the zoonosis that most affects humans worldwide and in the vast majority of the cases manifests itself as asymptomatic due to the immunocompetence of individuals [12].

• Percentages should be separated by dot instead of comma (Page 6, paragraph 3, line 138) E.g 65,9%

Line #132 - In Campos dos Goytacazes - RJ, studies were carried out with a sample of 367 individuals where 65.9 % of these showed positive serology for T. gondii [14].

• Reactional episodes in patients can occur before, during and after completing MDT, but you studied the co-parasitism with the occurrence of reactional episodes in patients before initiating MDT, why you wanted to do the study on before MDT? Do you have a reason? It would be good if you elaborate it. Page 7, paragraph 2, line 147.

The intention was to find a marker in an individual with leprosy that could be related to the development of a leprosy reaction. MDT administration could influence with immunological parameters. 

• Do you have a follow-up plan to study the effect of this co-parasitism with the occurrence of reactional episodes in patients during and after completion of MDT?

Yes. We have started to follow-up the individuals recently.

Methodology section :

• Would you define the Madrid classification? Since there are different types of leprosy classification, it would be better if you make it clear (Page 7, paragraph 3, line 157) 

We added in Line #152: The Madrid classification (1953) adopts polarity criteria, based on the clinical characteristics of the disease combined with the bacteriological criteria of leprosy. Thus, the polar groups are defined as: tuberculoid (T) and lepromatous (L); the transient and initial group of the disease, the indeterminate form (I); and the unstable and intermediate form, the borderline (B) or dimorphic (D) form (Souza, 1997).

• Neural leprosy patients were excluded from the analysis. Do you have specific reasons to exclude them from the study? (Page 7, paragraph 5, line 166)

There is no consensus regarding the classification of the neural form. In 1952, Wade classified it as an independent subgroup of leprosy, but this generated many discussions, as some authors believe that it is part of the subgroups of the disease. This form does not fit into any of the subgroups, so it must be analyzed separately, as it presents a difficult diagnosis, the clinical manifestations differ from the other forms, as it does not present skin lesions and the bacilloscopy is negative.

Please define the abbreviations:

 • Correct the ENL, Page 8, Figure 2, line 173: ENL: erythema nodosum leprosy

• anti-STAg, Page 8, Paragraph 2, line 178: anti-STAg (anti- Soluble Toxoplasma gondii Antigen) IgG

• The protein concentration in the supernatant (STAg) was determined by Lowry method. Would you please explain the method and also attach it as annex (page 9, Line 201).

We cited the reference (Lowry OH, Rosebrough NJ, Farr AL, Randall RJ. Protein measurement with the Folin phenol reagent. J Biol Chem. 1951;193:265–275) that we follow to measure the protein concentration of STAg.

Please define the abbreviations:

• PBST …Page 9, and Line 208: PBST (PBS [Phosphate Buffered Saline] 1×; 0.05% Tween 20). Then, 100 μl of blocking buffer (PBST, 1% BSA [Bovine Serum Albumin]) was added in each well and incubated for 30 min at 4 °C.

• BSA… Page 9, and Line 209: PBST (PBS [Phosphate Buffered Saline] 1×; 0.05% Tween 20). Then, 100 μl of blocking buffer (PBST, 1% BSA [Bovine Serum Albumin]) was added in each well and incubated for 30 min at 4 °C.

• I think Immunosorption written in the Enzyme Linked Immunosorption Assay (ELISA), should be changed to Immunosorbent assay Page 10, line 224

Yes. We change to the correct form in line #237 Enzyme Linked Immunosorbent Assay (ELISA) of immunoglobulins: IgA, IgE, IgG1, IgG2, IgG3 and IgG4.

Please define the abbreviation:

• ABTS stands for …Page 11, Paragraph 1, and Line 242: We inserted in line #227 (28 mmol/L citric acid, 48 mmol/L dehydrated sodium phosphate, 1 mg/ml ABTS [2,2'-AZINO-BIS(3-ETHYLBENZOTHIAZOLINE-6-SULFONIC ACID) DIAMMONIUM SALT] and 0.003% H2O2).

Results section:

• Percentages should be separated by dot instead of comma: OK. We changed all of them.

 • Page 11, Paragraph 2, line 262 and 263: OK, corrected.

• Check Table 2 summation: OK corrected

• Make corrections, highlighted Page 13, Line 282: OK. Corrected

Discussion:

• The cytokine TFN, Page 14, last paragraph, line 344: TNF-α

• The predictive value of anti-PGL-1 immunoglobulin levels for leprosy immune reactions is unclear. Please would you make the sentence clear? Which immunoglobulin? Are you referring to IgM or IgE levels? (Page 15, 1st paragraph, Line 347): Line #359 - There is no clear relationship between levels of any anti-PGL-1 immunoglobulins and predicted development of a leprosy reaction.

• Finally, high levels of TNF-a and NO mediated through IgE generation may also account for deleterious chronic inflammatory diseases that are observed during many parasitic infections as leprosy reactions. (Page 15, 1st paragraph, Line 363). The sentence is not clear it seems that leprosy reaction is a parasitic infection, though leprosy is a bacterial infection; leprosy reaction is not! Will you please rewrite it?: Line #376 - Finally, high levels of TNF-a and NO mediated through IgE generation may also account for deleterious chronic inflammatory diseases that are observed during many parasitic infections and which may be similar to what happens in leprosy reactions.

• Conclusion: I think this study is the ONLY study done on the increased risk of leprosy reaction with patients that are infected with T. gondii, so do you think it is possible to make conclusions as the presence of co-infection with T. gondii as high risk to develop leprosy reactions? Yes, so we rewrote the following to better explain: Our results suggest that T. gondii infections may play a role in the progression to more severe type of leprosy reaction, ENL, due to immunomodulation that influences the increase in IgE levels against the M. leprae antigen (PGL-1) in T. gondii coinfected patients, but not in those not coinfected.

 Reviewer #2:

Materials and Methods

• The authors should explain more clearly which groups are being studied and what the abbreviations mean. Line 163-166. The paragraph should be placed above on lines 156:

The Madrid classification (1953) adopts polarity criteria, based on the clinical characteristics of the disease combined with the bacteriological criteria of leprosy. Thus, the polar groups are defined as: tuberculoid (T) and lepromatous (L); the transient and initial group of the disease, the indeterminate form (I); and the unstable and intermediate form, the borderline (B) or dimorphic (D) form (Souza, 1997).

• Figure 2. Says study flow chart. There is an error it should say Figure 1: it was changed for Figure 1.

• Table 1. It does not provide further information. What is mentioned in the table could well go in the text: table 1 was removed and added the following text: Line #176 – From the total number of patients recruited, 180 (77 patients with leprosy reaction and 103 without leprosy reaction) were analyzed for the detection of anti-STAg (anti- Soluble Toxoplasma gondii Antigen) IgG antibodies and from these, 94 were analyzed regarding the immunoglobulin levels (IgA, IgE, IgG1, IgG2, IgG3 and IgG4) against anti-PGL-1 from M. leprae.

• Line 172. It says that not all participants were included in the laboratory studies. Describe how the selection was made, and what criteria were used:

The difference in the number of samples between the tests was due to the insufficient amount of reagents (antibodies) available to carry out the analysis of Igs anti-PGL-1 in all samples. The choice of samples was random for each clinical group. So we added in Line #180 - The difference in the number of samples between the tests was due to the insufficient amount of reagents (antibodies) available to carry out the analysis of Igs anti-PGL-1 in all samples. The choice of samples was random for each clinical group.

Results

Table 2. It is poorly tabulated and confusing. See another form of presentation: The Table 2 layout were changed in order promoting understanding.

Figure 2. The authors describe the results of the IGE experiment (lines 287-291), however there are other results that are not explained, such as, for example. IgG1 (values are high in both groups), IgG2 and 4 (large confidence intervals) will be that the groups correspond to a low number of patients, and IgG3 (the average value is high in both groups). All these results must be explained.

The values between the groups do not differ according to statistical analyses. In this sense, we add the following text: Line # 304 - There was no statistical significance in relation to the production levels of the immunoglobulins IgA, IgG1, IgG2, IgG3 and IgG4 between leprosy patients who developed leprosy reaction and those who did not develop a leprosy reaction (Figure 2).

Discussion

Line 329-333. The conclusion regarding M. leprae infection in the foot of the BALB/c mouse is not correct because the infection did not occur due to the immune protection of T. gondii but because the mouse is never susceptible to infection with Myocbacterium leprae. To affirm that, the authors mus look for a susceptible host like the armadillo.

Infection of the footpad of BALB/c nude mice with a viable M. leprae strain is a model used to maintain a viable M. leprae strain in laboratories. We ourselves make use of this M. leprae strain maintenance model in our laboratory. However, the authors of the cited work used an immunocompetent Balb/c mouse chronically infected with T. gondii to infect the footpad with a viable strain of M. leprae and observe the relationship of susceptibility and resistance related to the amount of M. leprae bacilli recovered from footpad. So, as this work referenced by us in the discussion was the only one in the scientific literature that showed a model of M. leprae/T. gondii co-infection, we use the author's conclusion in our discussion. The phrase “Contrary to our data, BALB/c mice chronically infected with the intracellular protozoan T. gondii or Besnoitia jellisoni were resistant to footpad challenge with M. leprae. Resistance was manifested by lower numbers of recoverable M. leprae in the footpads of protozoal-infected mice and resistance was enhanced in Toxoplasma-infected mice by a booster injection of Toxoplasma antigen in the infected footpad (Krahenbuhl JL, Levy L, Remington JS. Resistance to Mycobacterium leprae in Mice Infected with Toxoplasma gondii and Besnoitia jellisoni. Infect Immun. 1974 Nov;10(5):1068-71. doi: 10.1128/iai.10.5.1068-1071.1974. PMID: 16558091; PMCID: PMC423063.)” used in our discussion was not our conclusion, it was from the authors of the work we cited. We transcribe their conclusion in the context of our discussion even using the same words.

Comments

The authors make conclusions on a single biological marker. They should, as far as possible, associate the symptoms and clinical signs of each of the patients (Madrid Classification) with the respective markers and should monitor the marker (IgG E) over time at the time of detection and during treatment to effectively determine that immunomodulation is exerted.

We included in Line #391: Our results suggest that T. gondii infections may play a role in the progression to more severe type of leprosy reaction, ENL, due to immunomodulation that influences the increase in IgE levels against the M. leprae antigen (PGL-1) in T. gondii coinfected patients, but not in those not coinfected.

We also have added in Line #569 the following: 

Supporting information 

S1 File. Underlying Data Set. The spreadsheet called "Immunoglobulins" refers to the index values used in analyzing the levels of anti-PGL1 immunoglobulins. The worksheets called "Leprosy No Reaction", "Reverse Reaction". "Erythema Nodosum Leprosy" and "RR-ENL" present values used for risk analysis calculations.

---

## [Decision Letter · Decision Letter 1]

4 Jan 2024

PONE-D-23-32570R1Toxoplasma gondii infection and high levels of IgE are associated to erythema nodosum leprosy (ENL)PLOS ONE

Dear Dr. Rangel,

Thank you for submitting your manuscript to PLOS ONE. After careful consideration, we feel that it has merit but does not fully meet PLOS ONE’s publication criteria as it currently stands. Therefore, we invite you to submit a revised version of the manuscript that addresses the points raised during the review process.

**ACADEMIC EDITOR: ****Specifically, would you address or respond to reviewers comments related Madrid classification, editorial issues, and use of abbreviations.**==============================

We look forward to receiving your revised manuscript.

Kind regards,

Musa Mohammed Ali, PhD

Academic Editor

PLOS ONE

Journal Requirements:

Reviewers' comments:

Reviewer's Responses to Questions

**Comments to the Author**

1. If the authors have adequately addressed your comments raised in a previous round of review and you feel that this manuscript is now acceptable for publication, you may indicate that here to bypass the “Comments to the Author” section, enter your conflict of interest statement in the “Confidential to Editor” section, and submit your "Accept" recommendation.

Reviewer #1: All comments have been addressed

Reviewer #2: All comments have been addressed

2. Is the manuscript technically sound, and do the data support the conclusions?

Reviewer #1: Yes

Reviewer #2: Yes

3. Has the statistical analysis been performed appropriately and rigorously? 

Reviewer #1: Yes

Reviewer #2: Yes

4. Have the authors made all data underlying the findings in their manuscript fully available?

Reviewer #1: Yes

Reviewer #2: Yes

5. Is the manuscript presented in an intelligible fashion and written in standard English?

Reviewer #1: (No Response)

Reviewer #2: No

6. Review Comments to the Author

Reviewer #1: Few minor comments

I would like the authors to thank for incorporating the comments. I would appreciate if you include your responses to the manuscript so that readers will also understand the reason why you chose to do the study before MDT and also why you excluded the neural leprosy cases.

1. Page 4, Line 99 and 100

RR and ENL; please correct the abbreviations

Reverse Reaction (RR) to Reversal Reaction (RR)

Erythema Nodosum Leprous (ENR) to Erythema Nodosum Leprosum (ENL)

2. Please rewrite correctly the name of the chemical compound

Page 11, Paragraph 1, line 242

(2,2’ Azinobis acid; 3-ethylbenzothiazolin-6-sulfonic acid) diammonium salt (Sigma -Aldrich)

3. Page 13: Table 2 line 293

Please correct the abbreviations RR and ENL as mentioned above.

Reviewer #2: Materials and methods.

Line 151. Fig 1 must be placed in parentheses as in Line 160. (Fig 1),Line 301 (Fig 2) Line 304 (Fig 2), Line 306 (Fig 2b), Line 329 (Fig 3)

Line 152-157. The description of the Madrid classification is confusing. It is not only a histological classification but includes four aspects: clinical, baciloscopic, immunological and histopathological. It comprised two types Tuberculoid (T) and Lepromatous (L) and two unstable groups Indeterminate (I) and Domorphic or Borderline (D/B). (Souza, 1997). This cite must be added to the bibliopgraphy.

Line 171. Figure 1. Flow chart of study. LR: Leprosy Reaction; PB: paucibacillary leprosy without reaction; MB: multibacillary leprosy without reaction; RR: reversal reaction; ENL: erythema nodosum leprosy; RR/ENL two reactions at the same time; LR Before: patients diagnosed with reactions at the first clinic visit; LR After: patients that developed reaction after follow-up.

In this case, must be refered only as PB: paucibacillary clinical form and MB: multibacillary clinical form of the disease. (With reaction and without reaction must be eliminated, because those abbreviations are not in the figure). RR/ENL must be refered as: Reverse Reaction and Eritema Nodosum Leprosy that occurs simultaneously, in order to be consistent with the tables that follow.

Line 175. Correspond to the text of the Table 1 deleted. However the text is not clear enough regarding the number of patients analyzed.

From the total number of patients recruited, 180 (77 patients with leprosy reaction and 103 without leprosy reaction) were analyzed for the detection of anti-STAg (anti- Soluble Toxoplasma gondii Antigen) IgG antibodies and from these, 94 were analyzed regarding the immunoglobulin levels (IgA, IgE, IgG1, IgG2, IgG3 and IgG4) against anti-PGL-1 from M. leprae. The difference in the number of samples between the tests was due to the insufficient amount of reagents (antibodies) available to carry out the analysis of Igs anti-PGL-1 in all samples. The choice of samples was random for each clinical group.

This paragraph could be rewritten as: Of a 180 patients; 77 had leprosy reaction and 103 had not. Of this total, how many patients were checked with anti-STAg IgG ? in reference to the text that follows and which says “The difference in the number of samples between the tests was due to the insufficient amount of reagents (antibodies) available to carry out the analysis of Igs anti-PGL-1 in all samples” and that refers the current Table 1? . In this case the Table 1 must be added to the text.

Results.

Table 1. 1. You must justify the title, 2. tabulate columns 4 and 5, 3. add leprosy to column 1 (No Leprosy Reaction), 4. add the references of (RR) Reversal Reaction, (ENL) Erythema Nodosum leprosum, (RR/ENL) Reversal Reaction/Erythema Nodosum Leprosy that occur simultaneously. Although it may be redundant, in each of the tables the abbreviations at the footnotes must be clarified.

Table 2. At footnote must be added and the order of the words must be changed as (OR) Odds ratio, (CI) Confidential Interval, (RR) Reversal Reaction, (ENL) Erythema Nodosum leprosum, (RR/ENL) Reversal Reaction/Erythema Nodosum Leprosy that occur simultaneously.

Line 312. Figure 2. Immunoglobulins anti-PGL-1 present at (change at by in the) the serum from (change from by of) leprosy patients coinfected (Tx+) or not (Tx -) with T. gondii, who developed (with reaction) or not (without reaction) leprosy reactions. The IgA levels (A), (B) IgE, (C) IgG1, (D) IgG2, (E) IgG3 and (F) IgG4 were measured by ELISA and are presented as index value. The average of each patient were divided by the average of the

Figure 2. There was no statistical significance in relation to the production levels of the immunoglobulins between leprosy patients who developed leprosy reaction and those who did not develop a leprosy reaction Fig 2.

The paragraph could rewriting as: There was no significative differences in the production of IgA, IgG1, IgG2, IgG3 and IgG4 between leprosy and no leprosy patients.

Line 305. Nonetheless, leprosy patients who developed a leprosy reaction and…

The paragraph could rewrite as: Nonetheless, leprosy patients who developed leprosy reaction and…

Discusion

I think that rather than focusing on an experimental model, the authors should discuss other pathologies that co-exist with Leprosy infection such as Leishmanissis, tuberculosis, deep mycoses, etc. highly associated with leprosy given the social and economic conditions in which these populations co-habit.

7. PLOS authors have the option to publish the peer review history of their article (what does this mean?). If published, this will include your full peer review and any attached files.

Reviewer #1: No

Reviewer #2: No

---

## [Author Response · Author response to Decision Letter 1]

1 Feb 2024

Dear Plos One Academic Editor:

I am pleased to submit our revised manuscript entitled “Toxoplasma gondii infection and high levels of IgE are associated to erythema nodosum leprosy (ENL)” for publication as Research Article in the Plos One. Also, we would like to thank editor and reviewers for comments. Follow below our Response to Editor and Reviewers:

# Editor Comments:

All citation in the text was reviewed for reference list as required.

# Reviewer #1:

1) I would like the authors to thank for incorporating the comments. I would appreciate if you include your responses to the manuscript so that readers will also understand the reason why you chose to do the study before MDT and also why you excluded the neural leprosy cases.

We included in Line 170 the following: Patients with leprosy relapse and neural leprosy were excluded from the analysis due to there is no consensus regarding the classification of the neural form. In 1952, Wade classified it as an independent subgroup of leprosy, but this generated many discussions, as some authors believe that it is part of the subgroups of the disease. This form does not fit into any of the subgroups, so it must be analyzed separately, as it presents a difficult diagnosis, the clinical manifestations differ from the other forms, as it does not present skin lesions and the bacilloscopy is negative.

We incuded in Line 165 the following: “…since MDT administration could influence with immunological parameters.”

2) RR and ENL abbreviations page 4, paragraph 3, line 99 and 100.

Line 103 - They are divided into type 1 reaction, or Reversal Reaction (RR), and type 2 reaction, or Erythema Nodosum Leprosum (ENL), each one presenting specificity inherent to the pathophysiology, clinical picture and therapy [6].

3) Please rewrite correctly the name of the chemical compound Page 11, Paragraph 1, line 242 (2,2’ Azinobis acid; 3-ethylbenzothiazolin-6-sulfonic acid) diammonium salt (Sigma -Aldrich).

Line 257 - After this period, the wells were washed three times with Tris/Tween 0.05% 15mM (pH 7.5) and ABTS substrate solution (2,2’ Azinobis acid; 3-ethylbenzothiazolin-6-sulfonic acid) diammonium salt (Sigma -Aldrich) was added to reveal and kept in the dark for 15 to 20 minutes. The plates were read using a plate reader (EPOCH-BioTek) and the 405nm filter was used.

4) Page 13: Table 2 line 293 

Please correct the abbreviations RR and ENL as mentioned above.

Line 300: (OR) Odds ratio; (CI) Confidence Interval; (RR) Reversal Reaction; (ENL) Erythema Nodosum Leprosy; (RR/ENL) two reactions at the same time. Fisher's test for analyzes; *Statistical Significance.

# Reviewer #2: 

Materials and methods

1) Line 151. Fig 1 must be placed in parentheses as in Line 160. (Fig 1),Line 301 (Fig 2) Line 304 (Fig 2), Line 306 (Fig 2b), Line 329 (Fig 3)

All citation of figures was put in parenthesis.

2) Line 152-157. The description of the Madrid classification is confusing. It is not only a histological classification but includes four aspects: clinical, baciloscopic, immunological and histopathological. It comprised two types Tuberculoid (T) and Lepromatous (L) and two unstable groups Indeterminate (I) and Domorphic or Borderline (D/B). (Souza, 1997). This cite must be added to the bibliopgraphy.

Madrid classification, used in this work, does not use immunological and histopathological parameters. It uses only clinical and bacilloscopic parameters. The classification that uses immunological and histopathological parameters, beyond clinical and bacilloscopic, is the Ridley Jopling classification. So, in order to better explain the Madrid Classification, we added in Line 156 the following: The Madrid classification is based on clinical and bacilloscopic characteristics, dividing leprosy into two unstable groups, indeterminate and dimorphic or borderline, and two stable types, tuberculoid and lepromatous polar. 

3) Line 171. Figure 1. Flow chart of study. LR: Leprosy Reaction; PB: paucibacillary leprosy without reaction; MB: multibacillary leprosy without reaction; RR: reversal reaction; ENL: erythema nodosum leprosy; RR/ENL two reactions at the same time; LR Before: patients diagnosed with reactions at the first clinic visit; LR After: patients that developed reaction after follow-up.

In this case, must be refered only as PB: paucibacillary clinical form and MB: multibacillary clinical form of the disease. (With reaction and without reaction must be eliminated, because those abbreviations are not in the figure). RR/ENL must be refered as: Reverse Reaction and Eritema Nodosum Leprosy that occurs simultaneously, in order to be consistent with the tables that follow.

Line 176: Figure 1. Flow chart of study. LR: Leprosy Reaction; PB: paucibacillary; MB: multibacillary; RR: reversal reaction; ENL: erythema nodosum leprosy; RR/ENL two reactions at the same time; LR Before: patients diagnosed with reactions at the first clinic visit; LR After: patients that developed reaction after follow-up.

4) Line 175. Correspond to the text of the Table 1 deleted. However the text is not clear enough regarding the number of patients analyzed.

From the total number of patients recruited, 180 (77 patients with leprosy reaction and 103 without leprosy reaction) were analyzed for the detection of anti-STAg (anti- Soluble Toxoplasma gondii Antigen) IgG antibodies and from these, 94 were analyzed regarding the immunoglobulin levels (IgA, IgE, IgG1, IgG2, IgG3 and IgG4) against anti-PGL-1 from M. leprae. The difference in the number of samples between the tests was due to the insufficient amount of reagents (antibodies) available to carry out the analysis of Igs anti-PGL-1 in all samples. The choice of samples was random for each clinical group.

This paragraph could be rewritten as: Of a 180 patients; 77 had leprosy reaction and 103 had not. Of this total, how many patients were checked with anti-STAg IgG ? in reference to the text that follows and which says “The difference in the number of samples between the tests was due to the insufficient amount of reagents (antibodies) available to carry out the analysis of Igs anti-PGL-1 in all samples” and that refers the current Table 1? . In this case the Table 1 must be added to the text.

We changed in Line 180 the following: From 180 patients, 77 patients developed leprosy reaction and 103 not developed leprosy reaction. All of these patients were analyzed for the detection of anti-STAg (anti- Soluble Toxoplasma gondii Antigen) IgG antibodies. Of 180 patients, 94 were analyzed regarding the immunoglobulin levels (IgA, IgE, IgG1, IgG2, IgG3 and IgG4) against anti-PGL-1 from M. leprae (Table 1). The difference in the number of samples between the tests was due to the insufficient amount of reagents (antibodies) available to carry out the analysis of Igs anti-PGL-1 in all samples. The choice of samples was random for each clinical group.

We also included again the Table 1.

#Results.

5) Table 1. 1. You must justify the title, 2. tabulate columns 4 and 5, 3. add leprosy to column 1 (No Leprosy Reaction), 4. add the references of (RR) Reversal Reaction, (ENL)

OK. It was done.

6) Erythema Nodosum leprosum, (RR/ENL) Reversal Reaction/Erythema Nodosum Leprosy that occur simultaneously. Although it may be redundant, in each of the tables the abbreviations at the footnotes must be clarified.

OK. We changed all of these sugestion.

7) Table 2. At footnote must be added and the order of the words must be changed as (OR) Odds ratio, (CI) Confidential Interval, (RR) Reversal Reaction, (ENL) Erythema Nodosum leprosum, (RR/ENL) Reversal Reaction/Erythema Nodosum Leprosy that occur simultaneously.

Line 300: (OR) Odds ratio; (CI) Confidence Interval; (RR) Reversal Reaction; (ENL) Erythema Nodosum Leprosy; (RR/ENL) two reactions at the same time. Fisher's test for analyzes; *Statistical Significance.

8) Line 312. Figure 2. Immunoglobulins anti-PGL-1 present at (change at by in the) the serum from (change from by of) leprosy patients coinfected (Tx+) or not (Tx -) with T. gondii, who developed (with reaction) or not (without reaction) leprosy reactions. The IgA levels (A), (B) IgE, (C) IgG1, (D) IgG2, (E) IgG3 and (F) IgG4 were measured by ELISA and are presented as index value. The average of each patient were divided by the average of the

Line 318 - Figure 2. Immunoglobulins anti-PGL-1 present in the serum of leprosy patients coinfected (Tx+) or not (Tx -) with T. gondii, who developed (with reaction) or not (without reaction) leprosy reactions. The IgA levels (A), IgE (B), IgG1 (C), IgG2 (D), IgG3 (E) and IgG4 (F) were measured by ELISA and are presented as index value. The average of each patient were divided by the average of the blank of each ELISA plate in order to get an index value.

9) Line 305. Nonetheless, leprosy patients who developed a leprosy reaction and…

The paragraph could rewrite as: Nonetheless, leprosy patients who developed leprosy reaction and…

Line 309 - There was no significative differences in the production of IgA, IgG1, IgG2, IgG3 and IgG4 between those who developed leprosy reaction and those who did not develop a leprosy reaction (Fig 2). Nonetheless, leprosy patients who developed leprosy reaction and were also coinfected with T. gondii had elevated levels of IgE (Fig 2b)

#Discusion

10) I think that rather than focusing on an experimental model, the authors should discuss other pathologies that co-exist with Leprosy infection such as Leishmanissis, tuberculosis, deep mycoses, etc. highly associated with leprosy given the social and economic conditions in which these populations co-habit.

In fact, initially we also thought about writing in this sense, however these works address the issue of co-infection but do not provide much information about the pathological mechanisms. So, we wrote a little about other coinfection but, because we found immunoglobulin E as a marker for the development of leprosy reactions in individuals coinfected with T. gondii, we prioritized focusing our discussion on experimental models in which this marker were involved.

---

## [Decision Letter · Decision Letter 2]

20 Feb 2024

PONE-D-23-32570R2Toxoplasma gondii infection and high levels of IgE are associated to erythema nodosum leprosy (ENL)PLOS ONE

Dear Dr. Rangel,

Thank you for submitting your manuscript to PLOS ONE. After careful consideration, we feel that it has merit but does not fully meet PLOS ONE’s publication criteria as it currently stands. Therefore, we invite you to submit a revised version of the manuscript that addresses the points raised during the review process

We look forward to receiving your revised manuscript.

Kind regards,

Musa Mohammed Ali, PhD

Academic Editor

PLOS ONE

Journal Requirements:

Reviewers' comments:

Reviewer's Responses to Questions

**Comments to the Author**

1. If the authors have adequately addressed your comments raised in a previous round of review and you feel that this manuscript is now acceptable for publication, you may indicate that here to bypass the “Comments to the Author” section, enter your conflict of interest statement in the “Confidential to Editor” section, and submit your "Accept" recommendation.

Reviewer #1: All comments have been addressed

Reviewer #2: All comments have been addressed

2. Is the manuscript technically sound, and do the data support the conclusions?

Reviewer #1: Yes

Reviewer #2: Yes

3. Has the statistical analysis been performed appropriately and rigorously? 

Reviewer #1: Yes

Reviewer #2: Yes

4. Have the authors made all data underlying the findings in their manuscript fully available?

Reviewer #1: Yes

Reviewer #2: No

5. Is the manuscript presented in an intelligible fashion and written in standard English?

Reviewer #1: Yes

Reviewer #2: Yes

6. Review Comments to the Author

Reviewer #1: Last very minor comments

PONE-D-23-32570

I would like the authors to thank for incorporating the comments.

1. Page 13, Line 293 and line 306

Table 2 and 3

Please change Erythema Nodosum leprosy to Erythema Nodosum Leprosum (ENL)

Reviewer #2: line. 495. bibliography citation number 16 does not apply with the Madrid classification. You must change it

Table 2. The footnote must be single space

Regarding the Discussion. If the authors consider that it is not necessary to discuss the clinical aspects, I have to accept it although I consider that it could contribute an interesting aspect to the work.

7. PLOS authors have the option to publish the peer review history of their article (what does this mean?). If published, this will include your full peer review and any attached files.

Reviewer #1: No

Reviewer #2: No

---

## [Author Response · Author response to Decision Letter 2]

1 Mar 2024

Editor Comments:

All citation in the text was reviewed for reference list as required and we corrected the reference 14 because the authors name was duplicated as well as 2015 Tese de Doutorado. So, it is as following: 

14. Vieira, Flavia Pereira. Contaminação ambiental por oocistos de Toxoplasma gondii e toxoplasmose de veiculação hídrica sob a perspectiva da vulnerabilidade de aquíferos. 2015. Tese de Doutorado – Universidade Estadual do Norte Fluminense Darcy Ribeiro, Campos dos Goytacazes.

Reviewer #1:

1) Page 13, Line 293 and line 306

Table 2 and 3

Please change Erythema Nodosum leprosy to Erythema Nodosum Leprosum (ENL)

We changed it as required. 

Reviewer #2:

2) line. 495. bibliography citation number 16 does not apply with the Madrid classification. You must change it.

We changed the reference 16 in reference list for: 

16. Pavani RAB, Tonolli ER, D’Avila SCGP. Histopathological classification and clinical correlation of 50 leprosy cases from a Teaching Hospital, São José do Rio Preto, São Paulo state, Brazil. Medicina (Ribeirão Preto) 2008; 41 (2): 188-95.

3) Table 2. The footnote must be single space

OK. It was adjusted.

4) Regarding the Discussion. If the authors consider that it is not necessary to discuss the clinical aspects, I have to accept it although I consider that it could contribute an interesting aspect to the work.

We added a paragraph at the beginning of the discussion to contribute a little more about this aspect of co-infections and their clinical impact, as requested in the last review.

---

## [Editor Report · Decision Letter 3]

5 Mar 2024

Toxoplasma gondii infection and high levels of IgE are associated to erythema nodosum leprosy (ENL)

PONE-D-23-32570R3

Dear Dr. Rangel ,

We’re pleased to inform you that your manuscript has been judged scientifically suitable for publication and will be formally accepted for publication once it meets all outstanding technical requirements.

Kind regards,

Musa Mohammed Ali, PhD

Academic Editor

PLOS ONE
---

## [Editor Report · Acceptance letter]

23 Mar 2024

PONE-D-23-32570R3 

PLOS ONE

Dear Dr. Rangel, 

I'm pleased to inform you that your manuscript has been deemed suitable for publication in PLOS ONE. Congratulations! Your manuscript is now being handed over to our production team.

Kind regards, 

on behalf of

Dr. Musa Mohammed Ali 

Academic Editor

PLOS ONE